# An Online Sequence-to-Sequence Model Using Partial Conditioning

**Navdeep Jaitly**
Google Brain
ndjaitly@google.com

**David Sussillo**
Google Brain
sussillo@google.com

**Quoc V. Le**
Google Brain
qvl@google.com

**Oriol Vinyals**
Google DeepMind
vinyals@google.com

**Ilya Sutskever**
Open AI*
ilyasu@openai.com

**Samy Bengio**
Google Brain
bengio@google.com

## Abstract

Sequence-to-sequence models have achieved impressive results on various tasks. However, they are unsuitable for tasks that require incremental predictions to be made as more data arrives or tasks that have long input sequences and output sequences. This is because they generate an output sequence conditioned on an entire input sequence. In this paper, we present a *Neural Transducer* that can make incremental predictions as more input arrives, without redoing the entire computation. Unlike sequence-to-sequence models, the Neural Transducer computes the next-step distribution conditioned on the *partially* observed input sequence and the partially generated sequence. At each time step, the transducer can decide to emit zero to many output symbols. The data can be processed using an encoder and presented as input to the transducer. The discrete decision to emit a symbol at every time step makes it difficult to learn with conventional backpropagation. It is however possible to train the transducer by using a dynamic programming algorithm to generate target discrete decisions. Our experiments show that the Neural Transducer works well in settings where it is required to produce output predictions as data come in. We also find that the Neural Transducer performs well for long sequences even when attention mechanisms are not used.

## 1   Introduction

The recently introduced sequence-to-sequence model has shown success in many tasks that map sequences to sequences, e.g., translation, speech recognition, image captioning and dialogue modeling [17, 4, 1, 6, 3, 20, 18, 15, 19]. However, this method is unsuitable for tasks where it is important to produce outputs as the input sequence arrives. Speech recognition is an example of such an *online* task – users prefer seeing an ongoing transcription of speech over receiving it at the "end" of an utterance. Similarly, instant translation systems would be much more effective if audio was translated online, rather than after entire utterances. This limitation of the sequence-to-sequence model is due to the fact that output predictions are conditioned on the entire input sequence.

In this paper, we present a Neural Transducer, a more general class of sequence-to-sequence learning models. Neural Transducer can produce chunks of outputs (possibly of zero length) as blocks of inputs arrive - thus satisfying the condition of being "online" (see Figure 1(b) for an overview). The model generates outputs for each block by using a transducer RNN that implements a sequence-to-sequence model. The inputs to the transducer RNN come from two sources: the encoder RNN and its own

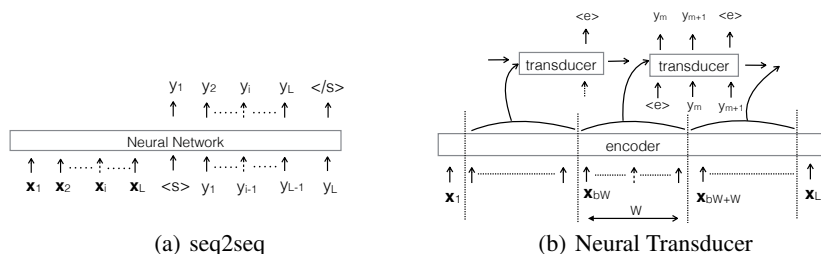

<div align="center">(a) seq2seq       (b) Neural Transducer</div>

Figure 1: High-level comparison of our method with sequence-to-sequence models. (a) Sequence-to-sequence model [17]. (b) The Neural Transducer (this paper) which emits output symbols as data come in (per block) and transfers the hidden state across blocks.

recurrent state. In other words, the transducer RNN generates local extensions to the output sequence, conditioned on the features computed for the block by an encoder RNN and the recurrent state of the transducer RNN at the last step of the previous block.

During training, alignments of output symbols to the input sequence are unavailable. One way of overcoming this limitation is to treat the alignment as a latent variable and to marginalize over all possible values of this alignment variable. Another approach is to generate alignments from a different algorithm, and train our model to maximize the probability of these alignments. Connectionist Temporal Classification (CTC) [7] follows the former strategy using a dynamic programming algorithm, that allows for easy marginalization over the unary potentials produced by a recurrent neural network (RNN). However, this is not possible in our model, since the neural network makes next-step predictions that are conditioned not just on the input data, but on the alignment, and the targets produced until the current step. In this paper, we show how a dynamic programming algorithm, can be used to compute "approximate" best alignments from this model. We show that training our model on these alignments leads to strong results.

On the TIMIT phoneme recognition task, a Neural Transducer (with 3 layered unidirectional LSTM encoder and 3 layered unidirectional LSTM transducer) can achieve an accuracy of 20.8% phoneme error rate (PER) which is close to state-of-the-art for unidirectional models. We show too that if good alignments are made available (e.g, from a GMM-HMM system), the model can achieve 19.8% PER.

## 2   Related Work

In the past few years, many proposals have been made to add more power or flexibility to neural networks, especially via the concept of augmented memory [10, 16, 21] or augmented arithmetic units [13, 14]. Our work is not concerned with memory or arithmetic components but it allows more flexibility in the model so that it can dynamically produce outputs as data come in.

Our work is related to traditional structured prediction methods, commonplace in speech recognition. The work bears similarity to HMM-DNN [11] and CTC [7] systems. An important aspect of these approaches is that the model makes predictions at every input time step. A weakness of these models is that they typically assume conditional independence between the predictions at each output step.

Sequence-to-sequence models represent a breakthrough where no such assumptions are made – the output sequence is generated by next step prediction, conditioning on the entire input sequence and the partial output sequence generated so far [5, 6, 3]. Figure 1(a) shows the high-level picture of this architecture. However, as can be seen from the figure, these models have a limitation in that they have to wait until the end of the speech utterance to start decoding. This property makes them unattractive for real time speech recognition and online translation. Bahdanau et. al. [2] attempt to rectify this for speech recognition by using a moving windowed attention, but they do not provide a mechanism to address the situation that arises when no output can be produced from the windowed segment of data.

Figure 1(b) shows the difference between our method and sequence-to-sequence models.

A strongly related model is the *sequence transducer* [8, 9]. This model augments the CTC model by combining the transcription model with a prediction model. The prediction model is akin to a

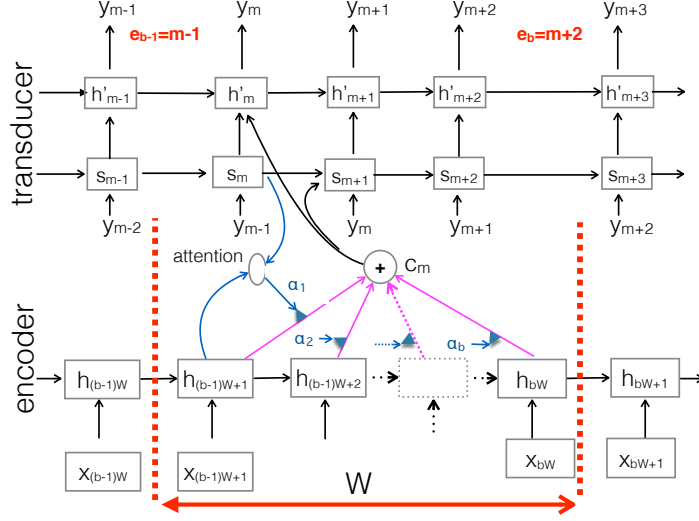

Figure 2: An overview of the Neural Transducer architecture for speech. The input acoustic sequence is processed by the encoder to produce hidden state vectors $\mathbf{h}_i$ at each time step $i$, $i = 1 \cdots L$. The transducer receives a block of inputs at each step and produces up to $M$ output tokens using the sequence-to-sequence model over this input. The transducer maintains its state across the blocks through the use of recurrent connections to the previous output time steps. The figure above shows the transducer producing tokens for block $b$. The subsequence emitted in this block is $y_m y_{m+1} y_{m+2}$.

language model and operates only on the output tokens, as a next step prediction model. This gives the model more expressiveness compared to CTC which makes independent predictions at every time step. However, unlike the model presented in this paper, the two models in the sequence transducer operate independently – the model does not provide a mechanism by which the prediction network features at one time step would change the transcription network features in the future, and vice versa. Our model, in effect both generalizes this model and the sequence to sequence model.

Our formulation requires inferring alignments during training. However, our results indicate that this can be done relatively fast, and with little loss of accuracy, even on a small dataset where no effort was made at regularization. Further, if alignments are given, as is easily done offline for various tasks, the model is able to train relatively fast, without this inference step.

## 3   Methods

In this section we describe the model in more detail. Please refer to Figure 2 for an overview.

### 3.1   Model

Let $\mathbf{x}_{1 \cdots L}$ be the input data that is $L$ time steps long, where $\mathbf{x}_i$ represents the features at input time step $i$. Let $W$ be the block size, i.e., the periodicity with which the transducer emits output tokens, and $N = \lceil \frac{L}{W} \rceil$ be the number of blocks.

Let $\tilde{y}_{1 \cdots S}$ be the target sequence, corresponding to the input sequence. Further, let the transducer produce a sequence of $k$ outputs, $\tilde{y}_{i \cdots (i+k)}$, where $0 \leq k < M$, for any input block. Each such sequence is padded with the $<e>$ symbol, that is added to the vocabulary. It signifies that the transducer may proceed and consume data from the next block. When no symbols are produced for a block, this symbol is akin to the blank symbol of CTC.

The sequence $\tilde{y}_{1 \cdots S}$ can be transduced from the input from various alignments. Let $\mathcal{Y}$ be the set of all alignments of the output sequence $\tilde{y}_{1 \cdots S}$ to the input blocks. Let $y_{1 \cdots (S+B))} \in \mathcal{Y}$ be any such alignment. Note that the length of $y$ is $B$ more than the length of $\tilde{y}$, since there are $B$ end of block symbols, $<e>$, in $y$. However, the number of sequences $y$ matching to $\tilde{y}$ is much larger, corresponding to all possible alignments of $\tilde{y}$ to the blocks. The block that element $y_i$ is aligned

to can be inferred simply by counting the number of <e> symbols that came before index $i$. Let, $e_b, b \in 1 \cdots N$ be the index of the last token in $y$ emitted in the $b^{th}$ block. Note that $e_0 = 0$ and $e_N = (S + B)$. Thus $y_{e_b} =$<e> for each block $b$.

In this section, we show how to compute $p\left(y_{1\cdots(S+B))}|\mathbf{x}_{1\cdots L}\right)$. Later, in section 3.5 we show how to compute, and maximize $p\left(\tilde{y}_{1\cdots S}|\mathbf{x}_{1\cdots L}\right)$.

We first compute the probability of l compute the probability of seeing output sequence $y_{1\cdots e_b}$ by the end of block $b$ as follows:

$$p\left(y_{1\cdots e_b}|\mathbf{x}_{1\cdots bW}\right) = p\left(y_{1\cdots e_1}|\mathbf{x}_{1\cdots W}\right) \prod_{b'=2}^{b} p\left(y_{(e_{b'-1}+1)\cdots e'_b}|\mathbf{x}_{1\cdots b'W}, y_{1\cdots e_{b'-1}}\right) \quad (1)$$

Each of the terms in this equation is itself computed by the chain rule decomposition, i.e., for any block $b$,

$$p\left(y_{(e_{b-1}+1)\cdots e_b}|\mathbf{x}_{1\cdots bW}, y_{1\cdots e_{b-1}}\right) = \prod_{m=e_{(b-1)}+1}^{e_b} p\left(y_m|\mathbf{x}_{1\cdots bW}, y_{1\cdots(m-1)}\right) \quad (2)$$

The next step probability terms, $p\left(y_m|\mathbf{x}_{1\cdots bW}, y_{1\cdots(m-1)}\right)$, in Equation 2 are computed by the transducer using the encoding of the input $\mathbf{x}_{1\cdots bW}$ computed by the encoder, and the label prefix $y_{1\cdots(m-1)}$ that was input into the transducer, at previous emission steps. We describe this in more detail in the next subsection.

## 3.2 Next Step Prediction

We again refer the reader to Figure 2 for this discussion. The example shows a transducer with two hidden layers, with units $\mathbf{s}_m$ and $\mathbf{h}'_m$ at output step $m$. In the figure, the next step prediction is shown for block $b$. For this block, the index of the first output symbol is $m = e_{b-1} + 1$, and the index of the last output symbol is $m + 2$ (i.e. $e_b = m + 2$).

The transducer computes the next step prediction, using parameters, $\theta$, of the neural network through the following sequence of steps:

$$\mathbf{s}_m = f_{RNN}\left(\mathbf{s}_{m-1}, [\mathbf{c}_{m-1}; y_{m-1}]; \theta\right) \quad (3)$$

$$\mathbf{c}_m = f_{context}\left(\mathbf{s}_m, \mathbf{h}_{((b-1)W+1)\cdots bW}; \theta\right) \quad (4)$$

$$\mathbf{h}'_m = f_{RNN}\left(\mathbf{h}'_{m-1}, [\mathbf{c}_m; s_m]; \theta\right) \quad (5)$$

$$p\left(y_m|\mathbf{x}_{1\cdots bW}, y_{1\cdots(m-1)}\right) = f_{softmax}\left(y_m; \mathbf{h}'_m, \theta\right) \quad (6)$$

where $f_{RNN}\left(\mathbf{a_{m-1}}, \mathbf{b_m}; \theta\right)$ is the recurrent neural network function (such as an LSTM or a sigmoid or tanh RNN) that computes the state vector $\mathbf{a}_m$ for a layer at a step using the recurrent state vector $\mathbf{a}_{m-1}$ at the last time step, and input $\mathbf{b}_m$ at the current time step;[2] $f_{softmax}\left(\cdot; \mathbf{a_m}; \theta\right)$ is the softmax distribution computed by a softmax layer, with input vector $\mathbf{a_m}$; and $f_{context}\left(\mathbf{s}_m, \mathbf{h}_{((b-1)W+1)\cdots bW}; \theta\right)$ is the context function, that computes the input to the transducer at output step $m$ from the state $s_m$ at the current time step, and the features $\mathbf{h}_{((b-1)W+1)\cdots bW}$ of the encoder for the current input block, $b$. We experimented with different ways of computing the context vector – with and without an attention mechanism. These are described subsequently in section 3.3.

Note that since the encoder is an RNN, $\mathbf{h}_{(b-1)W\cdots bW}$ is actually a function of the entire input, $\mathbf{x}_{1\cdots bW}$ so far. Correspondingly, $\mathbf{s}_m$ is a function of the labels emitted so far, and the entire input seen so far.[3] Similarly, $\mathbf{h}'_m$ is a function of the labels emitted so far and the entire input seen so far.

## 3.3 Computing $f_{context}$

We first describe how the context vector is computed by an attention model similar to earlier work [5, 1, 3]. We call this model the MLP-attention model.

In this model the context vector $\mathbf{c}_m$ is in computed in two steps - first a normalized attention vector $\alpha_m$ is computed from the state $\mathbf{s}_m$ of the transducer and next the hidden states $\mathbf{h}_{(b-1)W+1\cdots bW}$ of the encoder for the current block are linearly combined using $\alpha$ and used as the context vector. To compute $\alpha_m$, a multi-layer perceptron computes a scalar value, $e_j^m$ for each pair of transducer state $\mathbf{s}_m$ and encoder $\mathbf{h}_{(b-1)W+j}$. The attention vector is computed from the scalar values, $e_j^m$, $j = 1 \cdots W$. Formally:

$$e_j^m = f_{attention}\left(\mathbf{s}_m, \mathbf{h}_{(b-1)W+j}; \theta\right) \tag{7}$$

$$\alpha_m = softmax\left([e_1^m; e_2^m; \cdots e_W^m]\right) \tag{8}$$

$$\mathbf{c}_m = \sum_{j=1}^{W} \alpha_j^m \mathbf{h}_{(b-1)W+j} \tag{9}$$

We also experimented with using a simpler model for $f_{attention}$ that computed $e_j^m = \mathbf{s}_m^T \mathbf{h}_{(b-1)W+j}$. We refer to this model as DOT-attention model.

Both of these attention models have two shortcomings. Firstly there is no explicit mechanism that requires the attention model to move its focus forward, from one output time step to the next. Secondly, the energies computed as inputs to the softmax function, for different input frames $j$ are independent of each other at each time step, and thus cannot modulate (e.g., enhance or suppress) each other, other than through the softmax function. Chorowski et. al. [6] ameliorate the second problem by using a convolutional operator that affects the attention at one time step using the attention at the last time step.

We attempt to address these two shortcomings using a new attention mechanism. In this model, instead of feeding $[e_1^m; e_2^m; \cdots e_W^m]$ into a softmax, we feed them into a recurrent neural network with one hidden layer that outputs the softmax attention vector at each time step. Thus the model should be able to modulate the attention vector both within a time step and across time steps. This attention model is thus more general than the convolutional operator of Chorowski et. al. (2015), but it can only be applied to the case where the context window size is constant. We refer to this model as LSTM-attention.

### 3.4 Addressing End of Blocks

Since the model only produces a small sequence of output tokens in each block, we have to address the mechanism for shifting the transducer from one block to the next. We experimented with three distinct ways of doing this. In the first approach, we introduced no explicit mechanism for end-of-blocks, hoping that the transducer neural network would implicitly learn a model from the training data. In the second approach we added end-of-block symbols, <e>, to the label sequence to demarcate the end of blocks, and we added this symbol to the target dictionary. Thus the softmax function in Equation 6 implicitly learns to either emit a token, or to move the transducer forward to the next block. In the third approach, we model moving the transducer forward, using a separate logistic function of the attention vector. The target of the logistic function is 0 or 1 depending on whether the current step is the last step in the block or not.

### 3.5 Training

In this section we show how the Neural Transducer model can be trained.

The probability of the output sequence $\tilde{y}_{1..S}$, given $\mathbf{x}_{1\cdots L}$ is as follows[4]:

$$p\left(\tilde{y}_{1\cdots S}|\mathbf{x}_{1\cdots L}\right) = \sum_{y \in \mathcal{Y}} p\left(y_{1\cdots(S+B))}|\mathbf{x}_{1\cdots L}\right) \tag{10}$$

In theory, we can train the model by maximizing the log of equation 10. The gradient for the log likelihood can easily be expressed as follows:

$$\frac{d}{d\theta} \log p\left(\tilde{y}_{1\cdots S}|\mathbf{x}_{1\cdots L}\right) = \sum_{y \in \mathcal{Y}} p\left(y_{1\cdots(S+B))}|\mathbf{x}_{1\cdots L}, \tilde{y}_{1\cdots S}\right) \frac{d}{d\theta} \log p\left(y_{1\cdots(S+B)}|\mathbf{x}_{1\cdots L}\right) \tag{11}$$

Each of the latter term in the sum on the right hand side can be computed, by backpropagation, using $y$ as the target of the model. However, the marginalization is intractable because of the sum over a combinatorial number of alignments. Alternatively, the gradient can be approximated by sampling from the posterior distribution (i.e. $p\left(y_{1\cdots(S+B))}|\mathbf{x}_{1\cdots L}, \tilde{y}_{1\cdots S}\right)$). However, we found this had very large noise in the learning and the gradients were often too biased, leading to the models that rarely achieved decent accuracy.

Instead, we attempted to maximize the probability in equation 10 by computing the sum over only one term - corresponding to the $y_{1\cdots S}$ with the highest posterior probability. Unfortunately, even doing this exactly is computationally infeasible because the number of possible alignments is combinatorially large and the problem of finding the best alignment cannot be decomposed to easier subproblems. So we use an algorithm that finds the approximate best alignment with a dynamic programming-like algorithm that we describe in the next paragraph.

At each block, $b$, for each output position $j$, this algorithm keeps track of the approximate best hypothesis $h(j, b)$ that represents the best partial alignment of the input sequence $\tilde{y}_{1\cdots j}$ to the partial input $\mathbf{x}_{1\cdots bW}$. Each hypothesis, keeps track of the best alignment $y_{1\cdots(j+b)}$ that it represents, and the recurrent states of the decoder at the last time step, corresponding to this alignment. At block $b + 1$, all hypotheses $h(j, b), j <= \min\left(b\left(M - 1\right), S\right)$ are extended by at most $M$ tokens using their recurrent states, to compute $h(j, b + 1), h(j + 1, b + 1) \cdots h(j + M, b + 1)$[5]. For each position $j', j' <= \min\left((b + 1)\left(M - 1\right), S\right)$ the highest log probability hypothesis $h(j', b + 1)$ is kept[6]. The alignment from the best hypothesis $h(S, B)$ at the last block is used for training.

In theory, we need to compute the alignment for each sequence when it is trained, using the model parameters at that time. In practice, we batch the alignment inference steps, using parallel tasks, and cache these alignments. Thus alignments are computed less frequently than the model updates - typically every 100-300 sequences. This procedure has the flavor of experience replay from Deep Reinforcement learning work [12].

## 3.6   Inference

For inference, given the input acoustics $\mathbf{x}_{1\cdots L}$, and the model parameters, $\theta$, we find the sequence of labels $y_{1..M}$ that maximizes the probability of the labels, conditioned on the data, i.e.,

$$\tilde{y}_{1\cdots S} = \arg\max_{y_{1\cdots S'}, e_{1\cdots N}} \sum_{b=1}^{N} \log p\left(y_{e_{(b-1)+1}\cdots e_b}|\mathbf{x}_{1\cdots bW}, y_{1\cdots e_{(b-1)}}\right) \tag{12}$$

Exact inference in this scheme is computationally expensive because the expression for log probability does not permit decomposition into smaller terms that can be independently computed. Instead, each candidate, $y_{1\cdot S'}$, would have to be tested independently, and the best sequence over an exponentially large number of sequences would have to be discovered. Hence, we use a beam search heuristic to find the "best" set of candidates. To do this, at each output step $m$, we keep a heap of alternative $n$ best prefixes, and extend each one by one symbol, trying out all the possible alternative extensions, keeping only the best $n$ extensions. Included in the beam search is the act of moving the attention to the next input block. The beam search ends either when the sequence is longer than a pre-specified threshold, or when the end of token symbol is produced at the last block.

## 4   Experiments and Results

### 4.1   Addition Toy Task

We experimented with the Neural Transducer on the toy task of adding two three-digit decimal numbers. The second number is presented in the reverse order, and so is the target output. Thus the model can produce the first output as soon as the first digit of the second number is observed. The model is able to achieve 0% error on this task with a very small number of units (both encoder and transducer are 1 layer unidirectional LSTM RNNs with 100 units).

As can be seen below, the model learns to output the digits as soon as the required information is available. Occasionally the model waits an extra step to output its target symbol. We show results (blue) for four different examples (red). A block window size of W=1 was used, with M=8.

| 2 | + | 7 | 2 | 5 | \<s\> | 2 | 2 | 7 | + | 3 | \<s\> | |
|---|---|---|---|---|---|---|---|---|---|---|---|---|
| \<e\> | \<e\> | \<e\> | 9\<e\> | 2\<e\> | 5\<e\> | \<e\> | \<e\> | \<e\> | \<e\> | \<e\> | 032\<e\> | |
| 1 | 7 | 4 | + | 3 | \<s\> | 4 | 0 | + | 2 | 6 | 2 | \<s\> |
| \<e\> | \<e\> | \<e\> | \<e\> | \<e\> | 771\<e\> | \<e\> | \<e\> | \<e\> | \<e\> | 2\<e\> | 0\<e\> | 3\<e\> |

## 4.2 TIMIT

We used TIMIT, a standard benchmark for speech recognition, for our larger experiments. Log Mel filterbanks were computed every 10ms as inputs to the system. The targets were the 60 phones defined for the TIMIT dataset (*h#* were relabelled as *pau*).

We used stochastic gradient descent with momentum with a batch size of one utterance per training step. An initial learning rate of 0.05, and momentum of 0.9 was used. The learning rate was reduced by a factor of 0.5 every time the average log prob over the validation set decreased [7]. The decrease was applied for a maximum of 4 times. The models were trained for 50 epochs and the parameters from the epochs with the best dev set log prob were used for decoding.

We trained a Neural Transducer with three layer LSTM RNN coupled to a three LSTM layer unidirectional encoder RNN, and achieved a PER of 20.8% on the TIMIT test set. This model used the LSTM attention mechanism. Alignments were generated from a model that was updated after every 300 steps of Momentum updates. Interestingly, the alignments generated by the model are very similar to the alignments produced by a Gaussian Mixture Model-Hidden Markov Model (GMM-HMM) system that we trained using the Kaldi toolkit – even though the model was trained entirely discriminatively. The small differences in alignment correspond to an occasional phoneme emitted slightly later by our model, compared to the GMM-HMM system.

We also trained models using alignments generated from the GMM-HMM model trained on Kaldi. The frame level alignments from Kaldi were converted into block level alignments by assigning each phone in the sequence to the block it was last observed in. The same architecture model described above achieved an accuracy of 19.8% with these alignments.

For further exploratory experiments, we used the GMM-HMM alignments as given to avoid computing the best alignments. Table 1 shows a comparison of our method against a basic implementation of a sequence-to-sequence model that produces outputs for each block independent of the other blocks, and concatenates the produced sequences. Here, the sequence-to-sequence model produces the output conditioned on the state of the encoder at the end of the block. Both models used an encoder with two layers of 250 LSTM cells, without attention. The standard sequence-to-sequence model performs significantly worse than our model – the recurrent connections of the transducer across blocks are clearly helpful in improving the accuracy of the model.

Table 1: Impact of maintaining recurrent state of transducer across blocks on the PER (median of 3 runs). This table shows that maintaining the state of the transducer across blocks leads to much better results.

| W | BLOCK-RECURRENCE | PER |
|---|---|---|
| 15 | No | 34.3 |
| 15 | Yes | 20.6 |

Figure 3 shows the impact of block size on the accuracy of the different transducer variants that we used. See Section 3.3 for a description of the {DOT,MLP,LSTM}-attention models. All models used a two LSTM layer encoder and a two LSTM layer transducer. The model is sensitive to the choice of the block size, when no attention is used. However, it can be seen that with an appropriate choice of window size (W=8), the Neural Transducer without attention can match the accuracy of the attention based Neural Transducers. Further exploration of this configuration should lead to improved results.

When attention is used in the transducer, the precise value of the block size becomes less important. The LSTM-based attention model seems to be more consistent compared to the other attention

mechanisms we explored. Since this model performed best with W=25, we used this configuration for subsequent experiments.

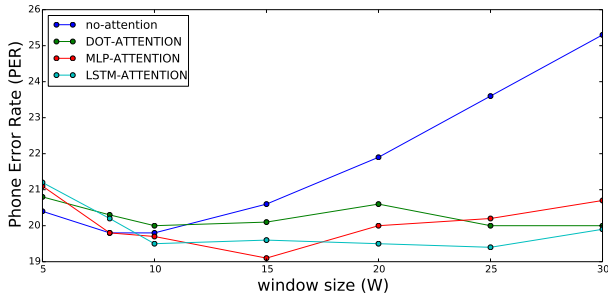

Figure 3: Impact of the number of frames (W) in a block and attention mechanism on PER. Each number is the median value from three experiments.

Table 2 explores the impact of the number of layers in the transducer and the encoder on the PER. A three layer encoder coupled to a three layer transducer performs best on average. Four layer transducers produced results with higher spread in accuracy – possibly because of the more difficult optimization involved. Thus, the best average PER we achieved (over 3 runs) was **19.8%** on the TIMIT test set. These results could probably be improved with other regularization techniques, as reported by [6] but we did not pursue those avenues in this paper.

Table 2: Impact of depth of encoder and transducer on PER.

| # of layers in encoder / transducer | 1 | 2 | 3 | 4 |
|---|---|---|---|---|
| 2 | | 19.2 | 18.9 | 18.8 |
| 3 | | 18.5 | **18.2** | 19.4 |

For a comparison with previously published sequence-to-sequence models on this task, we used a three layer bidirectional LSTM encoder with 250 LSTM cells in each direction and achieved a PER of 18.7%. By contrast, the best reported results using previous sequence-to-sequence models are 17.6% [6]. However, this requires controlling overfitting carefully.

## 5   Discussion

One of the important side-effects of our model using partial conditioning with a blocked transducer is that it naturally alleviates the problem of "losing attention" suffered by sequence-to-sequence models. Because of this, sequence-to-sequence models perform worse on longer utterances [6, 3]. This problem is automatically tackled in our model because each new block automatically shifts the attention monotonically forward. Within a block, the model learns to move attention forward from one step to the next, and the attention mechanism rarely suffers, because both the size of a block, and the number of output steps for a block are relatively small. As a result, error in attention in one block, has minimal impact on the predictions at subsequent blocks. Finally, we note that increasing the block size, $W$, so that it is as large as the input utterance makes the model similar to vanilla end-to-end models [5, 3].

## 6   Conclusion

We have introduced a new model that uses partial conditioning on inputs to generate output sequences. This allows the model to produce output as input arrives. This is useful for speech recognition systems and will also be crucial for future generations of online speech translation systems. Further it can be useful for performing transduction over long sequences – something that is possibly difficult for sequence-to-sequence models. We applied the model to a toy task of addition, and to a phone recognition task and showed that is can produce results comparable to the state of the art from sequence-to-sequence models.

## Footnotes

*Work done at Google Brain

[2]Note that for LSTM, we would have to additionally factor in cell states from the previous states - we have ignored this in the notation for purpose of clarity. The exact details are easily worked out.

[3]For the first output step of a block it includes only the input seen until the end of the last block.

[4]Note that this equation implicitly incorporates the prior for alignments within the equation

[5]Note the minutiae that each of these extensions ends with $<e>$ symbol.

[6]We also experimented with sampling from the extensions in proportion to the probabilities, but this did not always improve results.

[7]Note the TIMIT provides a validation set, called the *dev* set. We use these terms interchangeably.

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
