[Reviews · NeurIPS 2016]

Reviewer 1

Summary

The paper proposes a "neural transducer" model for sequence-to-sequence tasks that operates in a left-to-right and on-line fashion. In other words, the model produces output as the input is received instead of waiting until the full input is received like most sequence-to-sequence models do. Key ideas used to make the model work include a recurrent attention mechanism, the use of an end-of-block symbol in the output alphabet to indicate when the transducer should move to the next input block, and approximate algorithms based on dynamic programming and beam search for training and inference with the transducer model. Experiments on the TIMIT speech task show that the model works well and explore some of the design parameters of the model.

Qualitative Assessment

This is a well-done paper. It attacks a problem that is worthwhile: how to construct and train a sequence-to-sequence model that can operate on-line instead of waiting for an entire input to be received. It clearly describes an architecture for solving the problem, and walks the reader through the issues in the design of each component in the architecture: next-step prediction, the attention mechanism, and modeling the ends of blocks. It clearly explains the challenges that need to be overcome train the model and perform inference with it, and proposes reasonable approximate algorithms for training and inference. The speech recognition experiments used to demonstrate the utility of the transducer model and to explore design issues such as maintenance of recurrent state across block boundaries, block size, design of the attention mechanism, and depth of the model are reasonable. There are a few issues that should be addressed to improve the paper. Page 3, line 93: "We first compute the probability of l compute the probability of seeing output sequence" -- editing problem. Section 3.4: I was disappointed that there was no discussion of how well or poorly the three different approaches to end-of-block modeling worked. I assume that the use of the < e > symbol was best, but how much worse were the other two methods? Section 3.5: Computing alignments less frequently than model updates is also quite reminiscent of the use of lattices in sequence-discriminative training of acoustic models in speech recognition, as in D. Povey and P. C. Woodland, "Large scale discriminative training for speech recognition," in Proc. ASR Workshop, 2000, http://www.danielpovey.com/files/asr00.pdf, and B. Kingsbury, "Lattice-based optimization of sequence classification criteria for neural-network acoustic modeling," in Proc. ICASSP, 2009, https://www.semanticscholar.org/paper/Lattice-based-optimization-of-sequence-Kingsbury/2443dc59cf3d6cc1deba6d3220d61664b1a7eada/pdf In future work on the transducer model, it might be of interest to consider using lattices to represent alignment information. Methods developed for rescoring lattices using RNN language models might be useful for dealing with the fact that the neural transducer is conditioned on all of the input up to the present moment and all of the output labels generated up to the present moment. See, for example, X. Liu, Y. Wang, X. Chen, M. J. F. Gales, and P. C. Woodland, "Efficient lattice rescoring using recurrent neural network language models," in Proc. ICASSP, 2014, http://mi.eng.cam.ac.uk/projects/cued-rnnlm/papers/RNNLM_latrescore.pdf Page 7, lines 202-203: "Log Mel filterbanks" -> "Log Mel spectra" Section 4.2: It appears that the TIMIT experiments are not entirely standard. It is typical to collapse the TIMIT phones more than was done in this paper, and to specifically exclude certain data from the training set and to report results only on the core test set. See, for example T. N. Sainath, B. Ramabhadran, M. Picheny, D. Nahamoo, and D. Kanevsky, "Exemplar-Based Sparse Representation Features: From TIMIT to LVCSR," IEEE Transactions on Speech and Audio Processing, 19(8):2598--2613, Nov. 2011, https://36e9b848-a-62cb3a1a-s-sites.googlegroups.com/site/tsainath/tsainath_tsap2010_submission_2column_final.pdf or K. F. Lee and H. W. Hon, “Speaker-independent Phone Recognition Using Hidden Markov Models,” IEEE Transacations on Acoustics, Speech and Signal Processing, 37:1641–1648, 1989, http://repository.cmu.edu/cgi/viewcontent.cgi?article=2768&context=compsci The TIMIT experiments would be more easily compared to others if the standard framework were followed and this was clearly stated in the paper. Section 4.2: You should have tried initializing the transducer by training on the HMM-GMM alignments, and then continuing training with aligments inferred using the transducer. This might have led to even better TIMIT performance.

Confidence in this Review

3-Expert (read the paper in detail, know the area, quite certain of my opinion)


Reviewer 2

Summary

The authors propose a neural network method for learning to map input sequences to output sequences that is able to operate in an online fashion. Like similar models of this type, the input is processed by an encoder and a decoder produces an output sequence using the information provided by the encoder and conditioned on its own previous predictions. The method is evaluated on a toy problem and the TIMIT phoneme recognition task. The authors also propose some smaller ideas like two different attention mechanism variations.

Qualitative Assessment

I find the subject matter of the paper interesting as sequence/structured prediction with neural networks is still an open problem and I agree with the authors that there's a need for methods that can do online sequence processing and prediction. The biggest issue I have with the paper is that the term 'transducer' has been used multiple times before in the context of neural sequence prediction and the authors don't cite this work or discuss how it relates to their own method. This is somewhat surprising, because the authors do cite a paper in which both CTC and the transducer from Graves (2012) are evaluated and the transducer actually outperforms CTC. The transducer by Graves also takes into account previous predictions while still allowing for dynamic programming style inference as in CTC. This method differs from the method proposed by the authors and doesn't include the feature to process the input data in blocks but should be an important baseline to compare with. The existence of this prior work (and for example the work by Boulanger-Lewandowski et al., 2013) limits the novelty of the ideas presented in the paper and the extend to which the current title of the paper is appropriate. Perhaps something like "An Online Neural Transducer" or "An Incremental Neural Transducer" would shift the emphasis in the right direction. I find the empirical work somewhat limited. The results on TIMIT are decent, but don't really show the benefits of the new method in comparison to existing algorithms. I don't find the argument that it was beyond the scope of the paper to use regularization methods to get the best results possible convincing when the related work they compare with also uses an attention mechanism and windowing. That said, I did like the comparison with a model in which the RNN state is reset between different blocks and the plot about the relation between the window size and the use of attention mechanisms. All in all, it would have been interesting to see results on a dataset with longer sequences (like Wall Street Journal or Switchboard), where the new method may actually have a significant advantage.

Confidence in this Review

3-Expert (read the paper in detail, know the area, quite certain of my opinion)


Reviewer 3

Summary

The authors present a new sequence-to-sequence model architecture that processes fixed-size blocks of inputs using an encoder RNN and generates outputs block-by-block using a transducer RNN; the latter receives its inputs from the encoder RNN and maintains its state across blocks using recurrent connections. This class of models is aimed at online tasks like speech recognition that require predictions to be made incrementally as more data comes in.

Qualitative Assessment

This paper is well written and the idea of using a blocked transducer is novel. On the TIMIT core test set, the proposed online model performs quite comparably (19.8% PER) to the best-reported offline sequence-to-sequence model (17.6% PER), without using very carefully trained models. Since this model is designed to alleviate attention issues that particularly affect long utterances in sequence-to-sequence models, it would be interesting to see how the performance of the proposed model varies as a function of utterance length. One suggestion to make room for this experiment in the paper would be to drop the addition toy task which can be omitted without losing the flow of the narrative. There are a couple of missing implementation-specific details that’ll be useful for the reader to know: * In the comparison model (with a PER of 18.7%) that used a bidirectional encoder, were bidirectional features computed after seeing the end of an input block or after seeing the entire utterance? The latter would mean that the model is no longer offline. This should be clarified in the write-up. * What value of M was used in the experiments? (M appears in the DP algorithm used to compute alignments during training.) What beam search width was used during inference? As the authors have mentioned, the results reported in Table 2 could be improved by using better regularization techniques. Another way to potentially boost the performance of these models is to use better acoustic features (with speaker-dependent transforms). Lu et al., “Segmental Recurrent Neural Networks for End-to-end Speech Recognition”, Interspeech 2016 shows significant PER improvements on TIMIT obtained by using better acoustic features. When the authors say that the alignments from the proposed model are similar to the GMM-HMM alignments, do they mean this in terms of per-frame phone error rates? Clearly, the models benefited from using the GMM-HMM alignments as evidenced by the final PER numbers (19.8% vs 20.8%). What could this be attributed to, if the alignments were very similar? Some minor edits: — pg.3, typo in “compute the probability of 1 compute” — pg.4, “is in computed in” —> “is computed in” — pg.8, “showed that is can” —> “showed that it can” — argmax should be a single operator and not “arg max” in Eqn 12. — use \mathrm for the softmax function in Eqn 8.

Confidence in this Review

2-Confident (read it all; understood it all reasonably well)


Reviewer 4

Summary

This paper describes an online sequence-to-sequence model that emits output incrementally as it processes blocks of an input sequence. The map from block input to output is governed by a standard sequence-to-sequence model with additional state carried over from the previous block. Alignment of the two sequences is approximated by a dynamic program using a greedy local search heuristic. Experimental results are presented for phone recognition on TIMIT.

Qualitative Assessment

An incremental transducer is a natural extension of sequence-to-sequence models. Experimental results on TIMIT indicate that this model can be effectively optimized. I would guess that this model, or future iterations of it, will find lots of applications to data streams. A minor nitpick: the two tables in section 4.2 with two rows each seem a little tacky. Is there a better way to present this data? Also, column 1 of table 2 is empty.

Confidence in this Review

2-Confident (read it all; understood it all reasonably well)


Reviewer 5

Summary

The authors introduced an encoder-decoder architecture with attention over blocks of input and variable length decoder structure. The encoder is a multi-layer LSTM RNN. The decoder is an RNN model conditioned on weighted sums of the last layer of the encoder and it's previous output. The weighting schemes (attention) varies and can be conditioned on the hidden states or also previous attention vectors. The decoder model produces a sequence of symbols, until it outputs a special end character "e" and is moved to the next block (other mechanisms where explored as well (no end-of-block-symbol and separately predicting the end of a block given the attention vector). It is then fed the weighted sum of the next block of encoder states. The resulting sequence of symbols determines an alignment of the target symbols over the blocks of inputs, where each block may be assigned a variable number of characters. The system is trained by fixing an alignment, that approximately resembles the best alignment. Finding this approximately best alignment is akin to a beam-search with a beam size of M (line 169), but a restricted set of symbols conditional on the last symbol in a particular hypothesis (since the target sequence is known). Alignments are computed less frequently than model updates (typically every 100 to 300 sequences). For inference, an unconstrained beam-search procedure is performed with a threshold on sequence length and beam size.

Qualitative Assessment

First of all, I think the paper is well written and clear. Judging by the title the main contribution of this model is a novel model architecture. Looking at the references, there have previously been speech recognition systems, that use attention ([2] and [5]). Sequence transduction has also been introduced by Graves in "Sequence Transduction with Recurrent Neural Networks" in 2012. The follow up paper is cited as [7] uses this transduction method. Grave's describes his transducer to "extend CTC by defining a distribution over output sequences of all lengths, and by jointly modeling both input-output and output-output dependencies.". To me the novelty of this paper then is the idea of applying the transducer on a block-by-block basis and defining a RNN conditional on it's last definite output. Other methods integrate over alignments using the output of an RNN, whereas here the authors define a separate complex network structure to perform the transduction. However, Graves and Hinton also use a separate prediction network to feed into the transduction graph. To me, the disadvantage of the method presented here is the complicated training procedure, since exact constrained inference becomes intractable. I'm am also not sure how to judge the novelty of this approach, because I might be missing literature, since there has been a lot of developments in this area. It seems very similar to Graves and Hinton [7], especially since they also use the separate prediction network, except for the attention mechanism (but in later experiments we see, that the attention mechanism doesn't necessarily help that much (unless larger window sizes have advantages (one of which might be speed))). The experimental evaluation of this method is performed on an addition toy task and TIMIT. For the addition task the authors constrain themselves to adding two three-digit numbers. It would be interesting to see how the model generalizes to longer numbers. That is, if trained on additions of three-digit numbers, how well does it add four-digit numbers? For TIMIT the authors first compare having recurrent state across blocks, vs. not having a recurrent state. I think, if you don't have recurrent state in the transducer, you might be able to perform more efficient inference using Graves' transduction method. I'm not too sure on this point. Figure 3 shows that for some block sizes no attention performs equally well. I think no-attention means, that the encoder hidden states are averaged per block and then fed to the transducer. This is then akin to a convolution of size W and stride W. Maybe it is interesting to see how the encoder would perform, if W states are combined and fed into a simple RNN, that produces a softmax over symbols, which is then trained with CTC or Graves' Transducer. The best performance of Graves' Transducer on a unidirectional LSTM RNN has been reported at 19.6%, which is close to the best 19.8% reported here, albeit I understand, that the authors argue, that this model needs to be tuned more, which can be in itself a lot of work. Their best network uses a 3 layer LSTM RNN transducer. I think four layer transducers might vary much in accuracy, because they start overfitting a lot on TIMIT. It would be interesting to see how much these models overfit. Overall, given the complexity of the model and the performance, plus similarities to other methods, I'm not very sure how interesting this method is (I might lack the breath in literature to point out exactly how similar this model is). I would be interested in understanding how exactly it differs from Graves' and Hinton's Transducer network. I am not very sure on my review of the novelty of this method. The performance in itself is competitive on TIMIT, but other, simpler methods perform equally well. I am not sure if the extra capacity and context is necessary in comparison to other methods or for this dataset. Perhaps, if the model was trained on more data, it would become more apparent, but I'm not sure how well it scales. I wonder how long it takes to fully train a model?

Confidence in this Review

1-Less confident (might not have understood significant parts)